# Positive and Negative Affects and Cultural Attitudes among Representatives of the Host Population and Second-Generation Migrants in Russia and Kazakhstan

**Rail M. Shamionov** [1,*] **, Nasiya J. Sultaniyazova** [2] **and Alina S. Bolshakova** [3]

1   Department of Social Psychology of Education and Development, Saratov State University, Saratov 410012, Russia
2   Department of Pedagogy and Psychology, West-Kazakhstan Innovative-Technological University, Uralsk 090000, Kazakhstan
3   Department of English Language and Translation, Saratov State University, Saratov 410012, Russia
*   Correspondence: shamionov@mail.ru

**Abstract:** The experience of experiencing a new culture for first-generation migrants is usually quite an intense occurrence, one that has become the subject of numerous studies. However, the question of what happens later, at the level of the second and subsequent generations, is still under-investigated. The purpose of this study was to analyze the predictors of positive and negative affect of the second generation of migrants and representatives of the host (indigenous) population in Kazakhstan and Russia. The study involved 300 people selected on the basis of the principle of proportionality (quota selection). Survey methods and mathematical methods of data processing, including SEM (structural equation modeling), were used. The research model included comparative analysis of averages, regression analysis, and path analysis. The results testified in favor of the similarity of positive and negative affect indicators and their ratios in representatives of the host community and the second generation of migrants. Cultural attitudes of the host community representatives were characterized by higher certainty and rigidity than those of the representatives of the second generation of migrants. As a result of structural modeling, it was found that 20% of the positive affect dispersion in the representatives of the host community and 17% in the representatives of the second generation of migrants were conditioned by values, identity, and cultural attitudes. Positive affect in the representatives of the host community was associated with the values of self-overcoming, ethno-nihilism-based identity, and participation in cultural life of other peoples. Positive affect in representatives of the second generation of migrants was associated with the values of openness, attitude towards changing one's ethnic identity, positive attitude towards cultural borrowings, and a tendency to observe the traditions of one's ethnic group. Proposals have been formulated that contribute to reducing the cultural disunity of second-generation migrants and the host population.

**Keywords:** identities; values; affect; cultural attitudes; host community; second generation of migrants

## 1. Introduction

Migration processes have intensified globally in recent decades. Very often, this happens due to economic or environmental reasons, as well as military and political ones, which are common in different regions of the planet. It is no coincidence that researchers focus on the reasons for migratory movement, regardless of whether they are voluntary or involuntary. The decision-making patterns and motivational structures underlying migratory movement have been studied (Benet-Martínez and Repke 2020), and socio-psychological consequences of environmental influences on migrants have been established, including acculturation and adaptation patterns (Echterhoff et al. 2020). However, regardless of

reasons for migration, the problems of socio-psychological adaptation of migrants and representatives of the host community remain in the focus of researchers' attention.

Migration results not just in a change in "living conditions", but also changes one's basic identity and socio-cultural nature (Kuleshov 2007).

It is quite obvious that a person that undertakes migration actions a priori demonstrates a certain readiness to perceive a different culture and, if not to change their identity, then to transform it to a certain extent. Finnish researchers used regression analysis and conducted one of the most remarkable studies about the impact of the expected socio-cultural difficulties of socio-psychological adaptation involving the expected discrimination of Russian migrants and its impact on psychological well-being in the post-migration period. As a result of two studies carried out immediately before and after migration (in 2008 and 2010), they found that the expected socio-cultural difficulties of psychological adaptation mediate the perceived socio-cultural difficulties and stress of acculturation in the post-migration period; there is an impact of pre-acculturation stress and perceived discrimination on subsequent post-migration experiences (stress and discrimination, respectively), which are further associated with post-migration well-being. At the same time, it has been shown that migrants' adaptation is more favorable if there is a correlation between expected and real socio-cultural difficulties at a low level, with acculturation stress being lower than pre-acculturation stress, if they anticipate and perceive low level of discrimination (Mähönen and Jasinskaja-Lahti 2013). However, negative attitudes and behaviors towards immigrants often stem from people's ideologies that lead them to a certain vision of the world and intergroup relations (for example, orientation towards social dominance, authoritarianism, patriotism, nationalism), as well as psychosocial processes, such as formation of social identity (Esses 2020). For this reason, psychosocial research has also played an important role in identifying strategies to reduce prejudice and improve intergroup relations, in this case, in terms of relations between the host majority and immigrants (Cabaniss and Cameron 2018).

In the context of migrants' adaptation, understanding this process from the point of view of reciprocity becomes crucial, i.e., adaptation of both host community and migrants (Konstantinov 2017; Navas 2020; Drouhot and Nee 2019; Harpaz 2019). Therefore, recognition of the fact of changing identity and socio-cultural components in the migration environment and the host environment prove the necessity of studying socio-psychological characteristics in connection with basic psychological constructs in the interethnic environment.

Analysis of the processes of adaptation and factors leading migrants to well-being, which have been viewed in social and socio-cultural aspects, allows us to conclude that there are a number of changes in the system of values among representatives of ethnic groups (Azerbaijanis, Tajiks). For such groups, the development and acceptance of Russian reality and values becomes an adaptation factor. At the same time, acceptance of other values leads to weakening of the ethnic component of socio-cultural attitudes (Selyaninova 2020).

In the modern world, special efforts are being made to help migrants adapt to new socio-cultural conditions (Thomas et al. 2016; Martone et al. 2014; Nash et al. 2006). They have certain success in integrating migrants into the host community and sometimes even reduce the intensity of this process, reducing the risk of criminalization of migrant communities. Since adaptation experience of first-generation migrants is also important for socialization of their children, researchers have focused on studying ways to include parents and children in the culture of the host country (Barlow et al. 2016; Newland 2015; Eisenberg et al. 2001; Tsoukalas et al. 2010). Despite the experience of adaptation of migrants and refugees in various regions of the former Soviet Union, there are practically no studies on adaptation factors of the second and subsequent generations. One of the few exceptions is the analysis of the integration of second-generation migrants from the Transcaucasia and Central Asia into the Russian environment, undertaken by A.L. Rocheva, E.A. Varsaver, and N.S. Ivanova. Having explored identification aspects, these researchers came to the conclusion that under conditions of polyethnicity, ethnic identification ceases to be of decisive importance for some of the groups of migrants (Rocheva et al. 2019). Meanwhile,

preservation of collective historical memory and language of an ethnic group mediates the connection between ethnic identity and well-being (Lebedeva 2021). Under modern conditions of societies' digitalization, ethnic identity can be largely preserved through network interactions with the diaspora or ethnic group in the country of origin. Thus, involvement in virtual ethnic groups allows migrants to actualize their migration attitudes and preserve their ethnic identity (Melnikova et al. 2021).

Despite being born in the country, representatives of the second generation of migrants still experience certain difficulties. The results of the analysis of communicative interaction difficulties among migrant adolescents of the first and second generation in a cross-cultural aspect (Germany and Russia) revealed higher success of the second generation, and, consequently, a higher level of social adaptability. At the same time, difficulties of intercultural communication exist for both groups (Samokhvalova and Mets 2020).

Over the past two decades, empirical cross-cultural studies resulted in obtainment of data on the formation and transformation of subjective well-being, life satisfaction, and happiness of migrants under the influence of a complex of external and internal factors. In particular, studies of socio-psychological adaptation features of migrants in Russia have revealed a number of factors contributing to it (building social communication in a new environment, positive identity, self-actualization, etc.) (Gritsenko et al. 2021).

The study of migrants' subjective well-being made it possible to establish features of its structure and a number of factors of a socio-psychological nature that determine it in the migrant environment (social activity, positive ethnic identity, satisfaction with political activity, professional activity, etc. (Shamionov et al. 2013)). Influence of active position and motivation on the ability of migrants to overcome adaptation difficulties in a new socio-cultural environment has also been established, and, consequently, these factors have a positive impact on subjective well-being (Shamionov 2012).

At the same time, a significant share of the well-being/disadvantage of migrants is explained by the stress they experience in connection with migration. It is quite obvious that the second generation of migrants may indirectly experience psychological problems of their parents. M. Hynie's research evidences the significance of this stress for the mental health of migrants (Hynie 2018).

Finally, there are data obtained in interdisciplinary studies (correlations of socio-psychological views on migration with traditional sociological studies) on assimilation, stereotypes, and prejudices, as well as social identity, which are largely reflected in the mutual adaptation of migrants and members of the host population (Cabaniss and Cameron 2018) and also about semantic transformations in connection with social changes, as a result of migrations (Williamson et al. 2022).

In recent decades, studies of the longitudinal effects of migrants' acculturation and cultural stress have taken the most important place (Lorenzo-Blanco et al. 2017; Lorenzo-Blanco et al. 2016; Fuligni 2001). Meanwhile, "long" studies, studies of long-term socio-psychological effects of acculturation, and adaptation of migrants are promising in terms of revealing implicit problems in this process.

Despite great research interest in the problem of migration in sociology and social psychology, and despite the acuteness of the problem of adaptation of first-generation migrants, there are still questions about the basis of identification and cultural attitudes for adaptation of the second generation of migrants. This issue is of fundamental importance for many countries receiving migrants. Russia and Kazakhstan are such countries. In this sense, the study of second-generation migrants and representatives of the host population can be presented as a natural experiment in historically established conditions.

*Study Design*

The purpose of this study was to analyze the predictors of positive and negative affect of the second generation of migrants (settlers) and representatives of the host (indigenous) population in Kazakhstan and Russia.

The hypotheses

**H1.** *Representatives of the host population are characterized by more rigid attitudes towards the norms of ethnic groups than representatives of the second generation of migrants.*

**H2.** *Positive affectivity in the representatives of the second generation of migrants is conditioned by attitudes towards the norms of other ethnic groups. Negative affectivity is negatively associated with knowledge of the language of their ethnic group and participation in cultural life of other ethnic groups.*

**H3.** *The positive affect of the representatives of the host population is due to attitudes towards participation in the cultural life of other ethnic groups and positive acceptance of their ethnicity. The negative effect is associated with the emphasis on one's ethnicity and the attitude towards mixing cultures and traditions of ethnic groups.*

**H4.** *Positive affectivity in the representatives of the second generation of migrants is explained by the values of openness and attitude towards observance of the traditions of their ethnic group and, at the same time, positive attitude towards cultural borrowings. The positive affect of the representatives of the host population is explained by the value of self-overcoming, observance of the traditions of their people, and acceptance of their ethnicity.*

## 2. Materials and Methods

### 2.1. Participants

The study involved 300 people, including 150 representatives of the host population and 150 representatives of the second generation of migrants (settlers). Mean age—37.9 years, SD = 21.68; men—26%, women—74%, urban residents—69%, rural residents—31%. Table 1 presents socio-demographic scores of the host population and the second generation of migrants, as well as the general sample.

**Table 1.** Socio-demographic scores, scores of academic results, and health problems.

| | Host Population (n = 150) | | Second Generation of Migrants (n = 150) | | Total (n = 300) | |
|---|---|---|---|---|---|---|
| | %/Mean | SD | %/Mean | SD | %/Mean | SD |
| Age | 39.15 | (12.0) | 37.72 | (10.89) | 38.44 | (11.57) |
| Sex | | | | | | |
| Male | 25.3% | | 26.7% | | 26% | |
| Female | 74.7% | | 73.3% | | 74% | |
| Residence | | | | | | |
| Rural area | 26.7% | | 35.3% | | 34% | |
| City | 73.3% | | 64.7% | | 69% | |
| Education | | | | | | |
| School level | 11.4 | | 14.7 | | 13.0 | |
| Vocational | 27.5 | | 38.0 | | 32.8 | |
| University level | 61.1 | | 47.3 | | 54.2 | |

### 2.2. Method

To measure affect, we used the PANAS method (Positive and Negative Affect Schedule) (Crawford and Henry 2004) in cross-cultural adaptation by E. N. Osin (2012). When measuring affect (positive and negative), emotional states are reflected. At the same time, affects are also connected with personality traits that correspond to very stable individual differences in the tendency to display emotions of one type or another. We noticed a correlation between negative affect scores (irritability, nervousness, depression, etc.) and the experience of stress and frequency of negative life events (Watson et al. 1988). On the other hand, a correlation has been established between positive affect scores (confidence, cheerfulness, interest, etc.) and frequency of pleasant events, extraversive attitudes, social

activity, and even religious commitment (Watson 2002). The questionnaire consists of 20 adjectives characterizing various feelings and emotions. Respondents are asked to provide a score next to each adjective to indicate the degree of manifestation of these feelings in themselves during a certain period of time. In this case, we considered the period "as a whole".

The Schwartz PVQ-RR technique was used to measure value orientations: the manifestation degree for four value orientations was determined. Openness to Change Values emphasize willingness to embrace new or transformative ideas, actions, and experiences. Conservation Values emphasize avoidance of change, self-restraint, and order. Values of Self-Affirmation are included in the desire to satisfy one's own interests. Self-Overcoming Values are characterized by an emphasis on overcoming personal interests for the sake of others (Schwartz et al. 2012).

To assess ethnic identity, we used the "Types of Ethnic Identity" technique by G.U. Soldatova and S.V. Ryzhova. The respondent is offered stimulus material that consists of 30 positions that characterize their ethnic identity type. Respondents were asked to make a decision in relation to each of the positions in accordance with five response options, which are assigned the corresponding scores from 4 to 0: "agree" (4 points); "rather agree" (3 points); "somewhat agree, somewhat disagree" (2 points); "rather disagree" (1 point); "disagree" (0 points). Then, the number of points was calculated for each of the ethnic identity types: ethno-nihilism, ethnic indifference, norm (positive ethnic identity), ethno-egoism, ethno-isolationism, and ethno-fanaticism.

To assess major cultural attitudes, we used the authors' unique questionnaire called "Attitude to culture". It is aimed at assessing the attitude of respondents regarding support of ethno-cultural instructions, traditions, and norms. The questionnaire has a 5-point scale (1–5 points). For example: "Do you observe the traditions and customs of your people?" (1—no; 2—rather no than yes; 3—hard to answer; 4—rather yes than no; 5—yes, definitely).

A questionnaire was designed to identify socio-demographic indicators. Age, gender, religious commitment, place of residence, income level in the family, etc., were recorded.

All scores were tested for normal distribution using the Kolmogorov–Smirnov method. The scores of the Kolmogorov–Smirnov criterion provided a positive result ($p > 0.05$), which indicated the presumed correspondence of the empirical distributions to the normal one.

The average values of cultural attitudes of the two groups were obtained (the host population and representatives of the second generation of migrants). All the previous indicators met the requirements for the usage of this criterion. Then, we conducted a regression analysis by introducing affect indicators as a dependent variable, and cultural attitudes and types of ethnic identity as independent variables. In the next stage, we conducted a simulation procedure using AMOS for structural equation modeling. This program helped us to confirm the preliminary hypotheses and to establish the directions of relationship, and the criteria for model acceptance were set (chi-square (CMIN), degrees of freedom (df), comparative fit index (CFI), adjusted goodness-of-fit index (AGFI), goodness-of-fit index (GFI), and root mean square error of approximation (RMSEA)). In accordance with the model requirements (Hair et al. 2014), the statistical significance of all regression coefficients, and covariance between variables and variances were verified. Next, we analyzed the calculation results, and detected the direct and indirect effects and coefficient of determination ($R^2$) as a measure of the proportion of the variance of the dependent variable about its mean, which is explained by the independent variables (it is indicated in the image top right number on each dependent variable).

The statistical software IBM SPSS Statistics + PS IMAGO PRO (includes AMOS) was used for data processing.

## 3. Results

As can be seen from Table 2, there was a similarity in scores of following traditions, participation in the cultural life of one's own and other peoples, lack of desire to change ethnicity, and the tendency to limit a person's individuality by the moral and ethical norms

of an ethnic group. In other cases, there was a difference in the following scores: orientation in the customs of the ethnic group, use of set expressions of the ethnic group, interest in the art of the ethnic group, language proficiency, involvement in the culture of the ethnic group, the role of culture in ensuring unity of the ethnic group, tendency for reciprocal mixing of cultures of different ethnic groups, and tendency towards the development of the culture of an ethnic group due to diversity of cultures in the country.

**Table 2.** Cultural attitudes of the host population and representatives of the second generation of migrants.

|  | M1 | SD | M2 | SD | t | p |
|---|---|---|---|---|---|---|
| Tendency to follow ethnic group traditions (CA1) | 4.16 | 0.87 | 4.03 | 0.95 | 1.21 | 0.23 |
| Knowledge of ethnic traditions (CA10) | 4.09 | 0.81 | 3.78 | 0.93 | 3.13 | 0.00 |
| Tendency to use sayings (set expressions) of the ethnic group (CA2) | 4.08 | 1.10 | 3.57 | 1.33 | 3.60 | 0.00 |
| Ethnic language proficiency (CA8) | 4.30 | 1.00 | 3.81 | 1.31 | 3.62 | 0.00 |
| Tendency to participate in the people's religious life (CA3) | 4.38 | 0.77 | 4.25 | 0.85 | 1.42 | 0.16 |
| Inclusion in the culture and traditions of the ethnic group (CA11) | 4.01 | 1.05 | 3.43 | 1.06 | 4.75 | 0.00 |
| Desire to change ethnic identity (CA4) | 1.34 | 0.93 | 1.31 | 0.80 | 0.27 | 0.79 |
| Tendency to view the culture of one's people as a source of its unity and stability (CA12) | 4.25 | 0.94 | 3.79 | 1.06 | 3.98 | 0.00 |
| R) "Moral and ethical norms of an ethnos limit individuality" attitude (CA9) | 3.85 | 1.11 | 3.69 | 1.06 | 0.20 | 0.16 |
| Participation in the cultural life of other ethnic groups (CA5) | 3.44 | 1.50 | 3.65 | 1.27 | -1.33 | 0.19 |
| Positive attitude towards cultural borrowings (CA6) | 3.45 | 1.05 | 3.03 | 1.16 | 3.29 | 0.00 |
| Tendency to mix cultures and traditions (CA7) | 4.09 | 1.06 | 3.85 | 1.07 | 1.96 | 0.05 |
| R) Tendency to mix ethnic cultures (CA13) | 2.40 | 1.16 | 2.19 | 1.01 | 1.69 | 0.09 |
| Interest in the ethnic group's history (CA14) | 4.04 | 1.11 | 3.51 | 1.31 | 3.85 | 0.00 |

Note: R—reverse scale; M1—host population; M2—second generation of migrants.

Let us turn to regression analysis data concerning respondents' cultural attitudes, in which positive and negative affectivity was sequentially introduced as dependent variables. Table 3 shows that positive affectivity of the host population and the second generation of migrants was predicted by different cultural attitudes to varying degrees. Thus, they explained 13% of the variations in the positive affect of the host population and 16% of the positive affect of the representatives of the second generation of migrants.

**Table 3.** Cultural attitudes as predictors of positive affect in the host population.

| Cultural Attitudes | Beta | t | Value |
|---|---|---|---|
| **Host population** | | | |
| Participation in the cultural life of other ethnic groups (CA5) | 0.22 | 2.79 | 0.01 |
| Using sayings and proverbs of one's ethnic group in speech (CA2) | 0.19 | 2.35 | 0.02 |
| "Moral and ethical norms of an ethnos limit individuality" attitude (CA9) | 0.17 | 2.17 | 0.03 |
| | $R^2 = 0.13$, F = 7.1, $p < 0.001$ | | |
| **Representatives of the second generation of migrants** | | | |
| Tendency to follow ethnic group traditions (CA1) | 0.29 | 2.47 | 0.02 |
| Positive attitude towards cultural borrowing (CA6) | 0.16 | 2.09 | 0.04 |
| Participation in the religious life of one's ethnic group (CA3) | 0.19 | 2.32 | 0.02 |
| Desire to change ethnic identity (CA4) | 0.16 | 2.00 | 0.05 |
| | $R^2 = 0.16$, F = 6.95, $p < 0.001$ | | |

As can be seen from data presented in Table 4, 24% of the variations in the negative affect of the host population and only 10% of the variations in the negative affect of representatives of the second generation of migrants were explained by cultural attitudes.

**Table 4.** Cultural attitudes as predictors of negative affect in the host population.

| Cultural Attitudes | Beta | t | Value |
|---|---|---|---|
| **Host population** | | | |
| Moral and ethical norms of an ethnos limit individuality (CA9) | −0.19 | 2.48 | 0.01 |
| Interest in the history of an ethnic group (CA14) | −0.20 | 2.53 | 0.01 |
| R) Tendency to mix ethnic cultures (CA13) | 0.21 | 2.81 | 0.01 |
| Tendency to view the culture of one's people as a source of its unity and stability (CA12) | −0.24 | 2.82 | 0.01 |
| Participation in the religious life of one's ethnic group (CA3) | 0.20 | 2.65 | 0.01 |
| $R^2 = 0.24$, F = 9.1, $p < 0.001$ | | | |
| **Representatives of the second generation of migrants** | | | |
| Participation in the cultural life of other ethnic groups (CA5) | −0.26 | 3.25 | 0.001 |
| Ethnic language proficiency (CA8) | −0.16 | 2.03 | 0.04 |
| $R^2 = 0.10$, F = 8.2, $p < 0.001$ | | | |

Note: R—reverse scale.

To clarify the connection between the positive/negative affects experienced by the respondents, we carried out regression analysis, where ethnic identity types became independent variables. As can be seen from Table 5, positive and negative affectivity in the representatives of the host population was conditioned by ethno-nihilism, while negative affectivity in the representatives of the second generation of migrants was conditioned by ethno-egoism.

**Table 5.** Types of ethnic identity as predictors of positive/negative affect.

| | Beta | t | Value |
|---|---|---|---|
| **Positive affect** | | | |
| Host population: ethno-nihilism | −0.27 | 3.45 | 0.001 |
| $R^2 = 0.08$, F = 11.87, $p < 0.001$ | | | |
| **Negative affect** | | | |
| Host population: ethno-nihilism | 0.32 | 4.06 | 0.001 |
| $R^2 = 0.10$, F = 16.44, $p < 0.001$ | | | |
| Second generation of migrants: ethno-egoism | 0.29 | 3.61 | 0.001 |
| $R^2 = 0.08$, F = 13.04, $p < 0.001$ | | | |

We used structural equation modeling (SEM) to test the hypothesis on the direction of links to positive affect scores. As can be seen from the index scores of agreement placed under the figures, the presented models corresponded with the original data. In addition, all estimated parameters were statistically significant at $p < 0.05$. The model of the second generation of migrants (Figure 1) explained 17% of positive affect variations through the value of openness and three cultural attitudes: adherence to traditions, desire to change one's ethnic group, and recognition of the influence of other people's culture on the development of the culture of one's people. The host ethnic group model (Figure 2) explained 20% of positive affect variation. Among its predictors were ethno-nihilism ethnic identity type, the value of self-affirmation, and the orientation towards participation in the cultural life of other peoples.

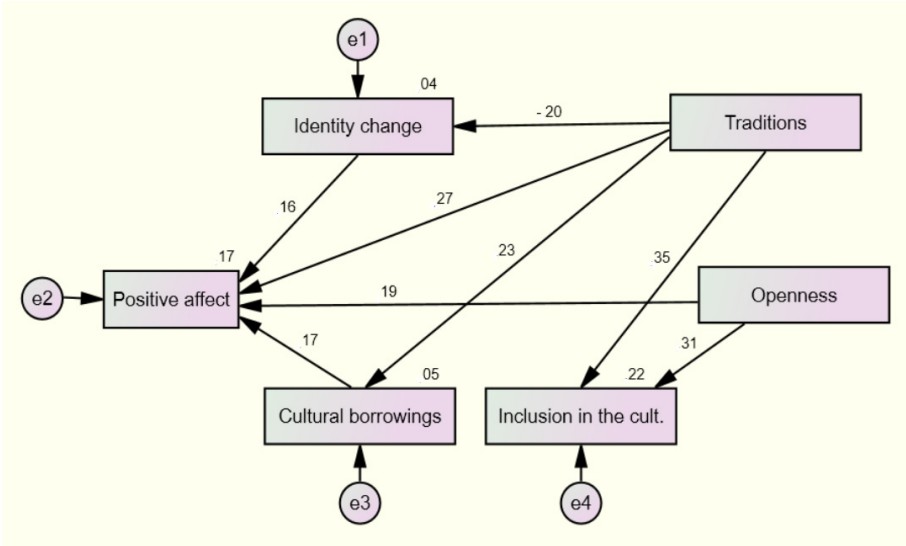

CMIN = 6.381; df = 7; *p* = 0.496; CFI = 1.000; AGFI = 0.959; GFI = 0.986; RMSEA = 0.000;
PCLOCE = 0.714

**Figure 1.** Model of migrants' cultural attitudes and positive affect paths.

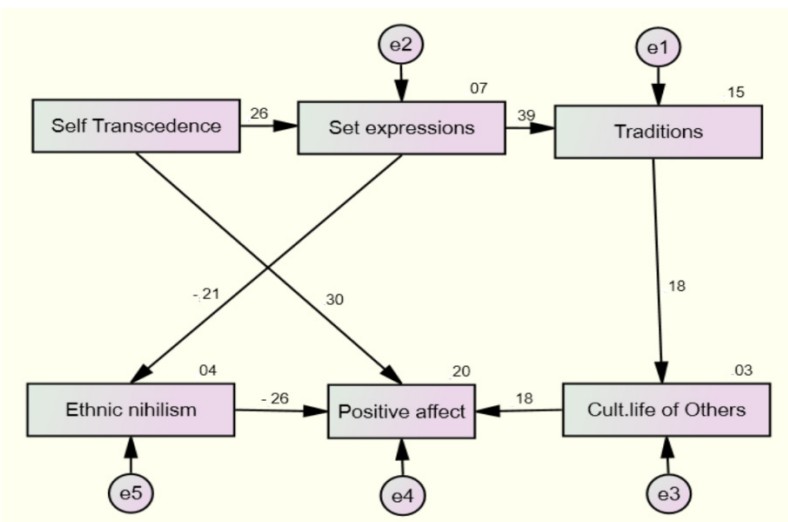

CMIN = 7.098; df = 84 *p* = 0.526; CFI = 1.000; AGFI = 0.961; GFI = 0.985;
RMSEA = 0.000; PCLOCE = 0.751

**Figure 2.** Model of host population's cultural attitudes and positive affect paths.

## 4. Discussion

Comparative analysis of the scores of cultural attitudes of the host population and representatives of the second generation of migrants resulted in the possibility of establishing similarities and differences in a number of important scores. We revealed similarly high scores for attitudes towards observing traditions of an ethnic group, participation in the cultural life of representatives of one's own and others' ethnic group, and unwillingness to change one's ethnic identity. These data are consistent with studies that show that representatives of the second generation of migrants in Russia are characterized by active life position while maintaining their ethno-cultural preferences as well as cultural ties with the diaspora (Kovaleva 2020). Meanwhile, higher scores were revealed among representatives of the host population for attitudes that ensure preservation of ethnic culture elements in everyday life: interest in the history of one's ethnic group, tendency to know the traditions of one's ethnic group, language, including the use of proverbs or sayings, recognition of

the role of culture in ethnic group preservation, expansion of culture to the level of other ethnic group cultures, and recognition of the tendency to mix cultures. The averages of the last two attitudes were in a lower position than the rest of the attitudes. These data testify in favor of higher level of ethnic consolidation of the host population. This is especially evident in the context of support for the policy of uniting all members of an ethnic group under a single citizenship or without it (Dumbrava 2019). Relatively low scores of adherence to ethnic culture among representatives of the second generation of migrants may indicate stronger involvement in the culture of the host ethnic group, as well as a less strong connection with cultural autonomy in a foreign cultural environment. Obviously, unlike the first generation of migrants, this environment is not alien and its representatives perceive the cultural phenomena of both their ethnic group and the host group. It is no coincidence that in this regard, they have a somewhat more positive attitude (at the level of a trend), in contrast to the host population, towards the mix of ethnic cultures (CA13). The situation of belonging to two or more cultures is not new and is often described in various sources, including modern studies, which point out the adaptive role of inclusion of the second generation of migrants in two or more cultures (Lee and Kim 2014; Schwartz et al. 2019). Despite similar socio-cultural conditions and similar newest history, the second generation of migrants in Russia and Kazakhstan profess different religions, which greatly increases cultural pressure and to a large extent prevents assimilation and promotes the search for culturally close people (and collective actions with representatives of one's ethnic group), which is consistent with data from the USA and Europe (Drouhot and Nee 2019). This requires special actions of states aimed not only at ensuring equal rights for representatives of ethnic groups but also actions for cultural integration and opportunities to realize the needs of the country's belonging to all its inhabitants.

Let us turn to data characterizing cultural attitude effects on the experience of positive and negative affect that characterize the degree of socio-psychological adaptation of the representatives of the two groups. As a result of regression analysis, where the affect scores were introduced as a dependent variable, and cultural attitude scores served as independent variables, it was possible to establish the contribution of attitudes to affect variations.

Positive affect in the representatives of the host community is explained by the joint action of a tendency to participate in the cultural life of other ethnic groups and the use of sayings and proverbs of one's own ethnic group in speech, which indicates a high degree of language proficiency and attitude towards the norms of the ethnic group as a behavior regulator. In other words, the combination of a positive attitude towards the norms of an ethnic group, language proficiency, and active participation in the cultural life of other peoples contributes to manifestation of a positive affect of the host population, which can serve as the basis for positive attitude towards representatives of other peoples. The positive affect of representatives of the second generation of migrants is explained by a combination of the attitude that culture of other peoples contributes to the development of culture of one's ethnic group, the tendency to participate in the cultural life of one's ethnic group, and the desire to change one's ethnic identity. These data indicate that positive affect of representatives of the second generation of migrants is largely conditioned by the integration and assimilation strategy, i.e., on the one hand, participation in the cultural life of one's own ethnic group, and on the other hand, positive attitude towards cultural borrowing and, albeit not at a high level, a certain willingness to change one's ethnic identity. To some extent, these data are consistent with the results of migrant assimilation studies (Drouhot and Nee 2019; Harpaz 2019), which noted a general trend of moving towards each other between migrants and the host population.

The negative affect of the representatives of the host population is positively explained by the tendency to mix cultures and traditions, as well as participation in the religious life of one's ethnic group, and negatively explained by the interest in the history of one's ethnic group, viewing one's culture as a source of its people's unity and stability, and the tendency to believe that moral and ethical norms of one's ethnic group do not limit one's individuality. These data testify in favor of the fact that the negative affect of the representatives of the

host population is associated with the dissonance between participation in the religious life of the people and the tendency to mix cultures and traditions of ethnic groups, reducing the negative affect of the tendency that characterizes a positive attitude towards the culture of one's ethnic group. The negative affect in representatives of the second generation of migrants is reduced by participation in the cultural life of other peoples and knowledge of the language of one's own ethnic group. In other words, intercultural interest is associated with a reduction in negative affect, which is consistent with the results of previous studies that established a causal relationship between interest and psychological adaptation of migrants (Gritsenko et al. 2021). These data confirm the significant role of adherence to the culture of one's ethnic group and interest in the culture of other peoples in the positive and negative affect variations. Meanwhile, positive affect is provided through a positive attitude towards the culture of one's own and another ethnic group, and the negative affect is provided only through characteristics of attitudes towards the culture of one's own ethnic group, usually of a derogatory nature. This may testify in favor of the fact that negative attitudes regarding participation in the cultural life of one's ethnic group, in the absence of a support group (ethnic group), lead to a general decrease in adaptation and, as a result, a negative affect.

Types of ethnic identity are weakly associated with affect scores. The positive and negative affect in the representatives of the host population are associated with ethno-nihilism identity type. The positive affect in the representatives of the host population is undermined by the ethno-nihilism identity type, which testifies in favor of the fact that the denial of inclusion in the formally assigned ethnic community of representatives of the host community contributes to negative affect and, accordingly, to psychological maladaptation. The negative affect of the representatives of the second generation of migrants is promoted by ethno-egoism identity type. These data are supported by the results of other studies, which confirm the connection between adaptation in the cultural environment of the majority with the ethnic indifference identification type (Konstantinov 2017).

Structural equation modeling (SEM) helped us to test the hypotheses about the paths between cultural attitudes and positive affect. On the basis of the obtained models, it follows that 20% of the variations in the positive affect in representatives of the host population was explained by the direct influence of the value of self-overcoming and ethno-nihilism identity type (negatively) and participation in the cultural life of other peoples, as well as with the moderating role of the attitude towards the use of sayings and proverbs of one's own ethnos in speech and observance of the traditions of one's people through the ethno-nihilism type, which lowers the causal link.

From the model of representatives of the second generation of migrants, it follows that 17% of the positive affect was conditioned by the combined action of the three attitudes and the value of openness. A direct causal relationship between attitudes towards observing traditions and positive affect was mediated by the attitude to change one's ethnic identity (negatively) and the attitude that the culture of other peoples contributes to the development of the culture of one's ethnic group, which characterizes migrants' positive attitude towards cultural borrowing from other peoples, or at least an interest in their culture. These data partially confirmed the results of studies that have established a relationship between intercultural competence and effectiveness of intercultural interactions (Khukhlaev et al. 2022). The link between positive affect and thoughts or even a desire to change one's ethnic identity may be conditioned by the well-known effect of demonstrating loyalty to the majority group in order to gain certain advantages (Harpaz 2019).

## 5. Conclusions

The study made it possible to establish that at the level of the second generation of migrants, the ratio of positive and negative affect does not differ from that of the representatives of the host population. However, different values, different types of ethnic identity, and different cultural attitudes are predictors of affect. Cultural attitudes of the representatives of the host community are characterized by higher certainty and rigidity

than those of the representatives of the second generation of migrants. The study also confirmed the view on the relationship between the positive affect of the second generation of migrants and openness to ethnic borrowings of the majority group, as well as the relation between positive affect of the representatives of the host community and orientation towards involvement in the culture of one's ethnic group. The practical conclusion of the study is the need for wider involvement of representatives of the second generation of migrants in civic participation and cultural life of the host population, which will reduce their cultural disunity.

## 6. Study Limitations

The limitations of the study were related to quantitative composition of the sample, the limitation in terms of representatives of ethnic groups (Kazakhs and Russians), and the sample geography (representatives of the border regions of the two countries that took part in the study). Subsequent research should clarify the issue concerning the depth of inclusion of first- and second-generation migrants in the culture of the host dominant ethnic group and in mutual relations. Previously, an indirect positive effect of civic identity and identity with a place on self-esteem was established through an integration strategy using the example of migrants in Lithuania and Georgia (Ryabichenko et al. 2019). Interestingly, identity with the place of residence performs the function of integrating an ethnic group in a foreign cultural context and serves to preserve the group's ethnic identity and successful adaptation. Obviously, the acceptance of one's stay in a foreign cultural environment plays an important role in adaptation while maintaining one's ethnicity. Therefore, an important circumstance is tracking the ethnic and national identity of migrants over time at the level of intergenerational cross-sections. Subsequent studies will be devoted to this issue.

**Author Contributions:** Conceptualization, R.M.S. and N.J.S.; methodology, N.J.S.; software, N.J.S.; validation, R.M.S., N.J.S. and A.S.B.; formal analysis, R.M.S. and N.J.S.; investigation, R.M.S.; data curation, N.J.S.; writing—original draft preparation, R.M.S.; writing—review and editing, N.J.S. and A.S.B.; visualization, R.M.S.; supervision, R.M.S.; project administration, R.M.S.. All authors have read and agreed to the published version of the manuscript.

**Funding:** This research received no external funding.

**Informed Consent Statement:** Informed consent was obtained from all subjects involved in the study.

**Data Availability Statement:** The data presented in this study are available on request from the corresponding author.

**Conflicts of Interest:** The authors declare no conflict of interest.

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
