# Peer review of "Positive and Negative Affects and Cultural Attitudes among Representatives of the Host Population and Second-Generation Migrants in Russia and Kazakhstan"

_socsci, doi:10.3390/socsci11100473_

Round 1
Reviewer 1 Report
Dear authors, I consider your research subject interesting. Especially, you use Structural equation modeling (SEM) and it is very useful. The research is well-structured and the integrity of the manuscript is well.
In the abstract section, you should mention your research model and sampling method and also add a suggestion sentence.
You should make clarification and justification why you choose two cases (Russia and Kazakhstan).
Your results are useful for migration studies, but you focused on only the positive aspects, it would be great if you could also show the negative aspects of immigration in Russia and Kazakhstan. Maybe this aspect is not directly related to your research purpose. It's your choice!
In the Conclusions section, I couldn't see your research limitations and practical implicates. What does say your study to migrations studies? What kind of contributions does make it? You should explain these.
Your references are a little narrow and they focus on a certain region. You should prefer other current studies around the world. I recommend you Michaela Hynie's and Derya Ozkul's studies about migration.
Author Response
First of all we would like to thank the reviewer for the useful suggestions. All the revisions on the manuscript were clearly highlighted (in red). The replies to the indications of the reviewer can be found in the table below on the right-hand side.

Reviewer 2 Report
The paper explores an important topic of the predictors of positive and negative affect in the still unresearched second generation of immigrants. The paper itself contributes to the literature, yet, in its current form, the paper is hard to read and inconsistent.
1. The paper's title suggests research on the positive affect and attitudes among second-generation migrants in Russia and Kazakhstan. Yet, the paper explores this topic in both migrants and host communities and compares cultural attitudes between these groups. You have two options to resolve this inconsistency. First, the paper can follow the title and focus only on second-generation migrants. This would mean removing the results of the host population from the paper. Second, which will require much more work, theoretically introducing why it is important to study the host society's attitudes and why it is necessary to compare them to the attitudes of second-generation immigrants. This would also require developing hypotheses about it. My to-go would be the first option, as the real contribution of the paper is research on the second generation of immigrants.
2. Regarding the analysis, it is not clear how it is done. To test your current hypotheses about predictions, two simple regression analyses should be done and only with second-generation migrants. In those regressions, you would enter all potential predictors together and a) positive affect and b) negative affect as the outcome variable. No further analysis should be done.
I do not see a need for SEM. Also, how is SEM done? Does SPSS have the possibility of conducting SEM? Also, 150 participants are not enough for SEM. Did you do path analysis? Finally, in case you decide to present SEM analysis, please elaborate on which hypothesis it answers, and please adapt SEM figures to contain names and not codes.
Author Response

(The authors gave the same response as above.)

Round 2
Reviewer 2 Report
Thank you for addressing my comments.
Good luck with the future work!